# Impact of Intravenous Iron Substitution on Serum Phosphate Levels and Bone Turnover Markers—An Open-Label Pilot Study

**DOI:** 10.3390/nu15122693

**Published:** 2023-06-09

**Authors:** Alexandra Struppe, Jakob E. Schanda, Andreas Baierl, Paul Watzl, Christian Muschitz

**Affiliations:** 1St. Vincent Hospital Vienna, II Medical Department-Gastroenterology, Hepatology, Metabolic, and Inflammation Medicine, Academic Teaching Hospital of the Medical University of Vienna, 1060 Vienna, Austria; alexandra.struppe@gmx.at (A.S.); paul.watzl@bhs.at (P.W.); 2AUVA Trauma Center Vienna-Meidling, Department for Trauma Surgery, 1120 Vienna, Austria; jakob.schanda@auva.at; 3Ludwig Boltzmann Institute for Traumatology, 1200 Vienna, Austria; 4Austrian Cluster for Tissue Regeneration, 1200 Vienna, Austria; 5Insitute for Statistics and Operations Research, University of Vienna, 1090 Vienna, Austria; andreas.baierl@univie.ac.at; 6HealthPi Medical Center, 1010 Vienna, Austria

**Keywords:** iron substitution therapy, ferric carboxymaltose, iron sucrose, phosphate, hypophosphatemia, 25-hydroxyvitamin D (25(OH)D), bone turnover markers, procollagen type 1 amino-terminal propeptide (P1NP), beta-CrossLaps (CTX)

## Abstract

The association between intravenous iron substitution therapy and hypophosphatemia was previously reported in patients with iron deficiency anemia. However, the extent of hypophosphatemia is thought to depend on the type of iron supplementation. We hypothesized that the intravenous application of ferric carboxymaltose and iron sucrose leads to a different longitudinal adaptation in serum phosphate levels. In this open-label pilot study, a total of 20 patients with inflammatory bowel diseases or iron deficiency anemia were randomly assigned to one of two study groups (group 1: ferric carboxymaltose, *n* = 10; group 2: iron sucrose, *n* = 10). Serum values were controlled before iron substitution therapy, as well as 2, 4, and 12 weeks after the last drug administration. The primary objective of the study was the longitudinal evaluation of serum phosphate levels after iron substitution therapy with ferric carboxymaltose and iron sucrose. The secondary objective was the longitudinal investigation of calcium, 25-hydroxyvitamin D (25(OH)D), intact parathyroid hormone, procollagen type 1 amino-terminal propeptide (P1NP), beta-CrossLaps (CTX), hemoglobin (Hb), iron, ferritin, and transferrin saturation levels. Two weeks after drug administration, phosphate levels were significantly lower (*p* < 0.001) in group 1 and ferritin levels were significantly higher (*p* < 0.001) in group 1. Phosphate levels (0.8–1.45 mmol/L) were below the therapeutic threshold and ferritin levels (10–200 ng/mL for women and 30–300 ng/mL for men) were above the therapeutic threshold in group 1. P1NP (15–59 µg/L) and CTX (<0.57 ng/mL) levels were above the therapeutic threshold in group 2. Four weeks after drug administration, significant differences were still observed between both study groups for phosphate (*p* = 0.043) and ferritin (*p* = 0.0009). All serum values except for Hb were within the therapeutic thresholds. Twelve weeks after drug administration, no differences were observed in all serum values between both study groups. Hb values were within the therapeutic threshold in both study groups. Serum 25(OH)D levels did not differ between both study groups throughout the whole study period and remained within the therapeutic threshold.

## 1. Introduction

The association between intravenous iron substitution therapy and hypophosphatemia was first reported in 1983 [1]. Shimizu et al. emphasized the underlying association between sugared iron oxide and hypophosphatemia and described a disorder of renal phosphate resorption, a low 25-hydroxyvitamin D (25(OH)D) blood concentration, and elevated concentrations of fibroblast growth factor 23 (FGF23) [2]. The reason for transient hypophosphatemia after intravenous iron substitution therapy is renal phosphate loss caused by iron-induced FGF23 stabilization, secondary hyperparathyroidism, and 25(OH)D resistance [3].

Chronic hypophosphatemia can affect bone metabolism and cause changes in bone structure and microarchitecture, eventually leading to osteomalacia and bone fragility [3,4]. Bartko et al. reported a case of a patient with Crohn’s Disease who developed hypophosphatemia and deficiency fractures during intravenous iron substitution therapy. A bone biopsy revealed a significant decrease in trabecular bone volume and thickness compared to the age-identical reference range. On the other hand, the osteoid volume was nine times higher, while the osteoid surface area and the osteoid thickness were four times higher [4].

As the extent of hypophosphatemia is thought to be dependent on the type of iron supplementation [1], the severity of hypophosphatemia and concomitant osteomalacia may vary according to different intravenous iron substitution regimes. We hypothesized that the intravenous application of ferric carboxymaltose and iron sucrose lead to a different longitudinal adaptation in serum phosphate levels. The primary objective of this study was the longitudinal evaluation of serum phosphate levels after iron substitution therapy with ferric carboxymaltose and iron sucrose. The secondary objective was the longitudinal investigation of calcium, 25(OH)D, intact parathyroid hormone, procollagen type 1 amino-terminal propeptide (P1NP), beta-CrossLaps (CTX), hemoglobin (Hb), iron, ferritin, and transferrin saturation levels after iron substitution therapy with ferric carboxymaltose and iron sucrose.

## 2. Materials and Methods

The study was performed in accordance with the Helsinki Declaration and was authorized and controlled by the responsible ethics committee of St. Vincent Hospital Vienna (no. EK20/2018). Each patient signed written informed consent prior to any study procedures. The study was performed at the II Medical Department of Gastroenterology, Hepatology, Metabolic, and Inflammation Medicine at the St. Vincent Hospital of Vienna. The investigation was set as a parallel-group study with a randomized, monocentric, prospective, controlled, and unblinded design.

In this open-label pilot study, a total of 20 patients were randomly assigned to one of the two study groups (group 1: ferric carboxymaltose, *n* = 10; group 2: iron sucrose, *n* = 10). Inclusion criteria were an underlying chronic inflammatory bowel disease (Crohn’s Disease, ulcerative colitis) with occult blood loss determined by a fecal test, or another underlying disease requiring intravenous iron substitution such as iron deficiency anemia. The diagnosis was based on serum levels presenting with microcytic hypochromic anemia or a fecal test. Patients were excluded in cases of underlying oncological disease, ongoing oral iron substitution therapy, or acute bleeding anemia. Furthermore, postoperative patients as well as patients requiring intensive care were excluded from the study. Additionally, patients with nutritional deficiencies due to food restrictions (e.g., veganism), chronic blood loss in premenopausal women, or malabsorption due to bariatric surgery were excluded from the study. Patients were further instructed not to receive any other treatments during the study period besides their ongoing long-term medication, as this could have affected the study outcome. Additionally, patients were advised not to undergo any dietetical changes or adaptations to their medication. Group 1 received one single dose of 1000 mg of ferric carboxymaltose (Ferinject^®^, CSL Vifor, St. Gallen, Switzerland) intravenously. Group 2 was treated once daily with 200 mg of iron sucrose (FerMed^®^, Medice Arzneimittel GmbH, Hallein, Austria) intravenously for three consecutive days.

Blood samples were collected initially before iron substitution, as well as 2, 4, and 12 weeks after the last intravenous iron administration. At the time of reappointment, fasting blood probes were obtained from all patients between 8:00 and 10:00. These were instantly centrifuged, refrigerated, and kept at −70 °C for further analysis. All laboratory analyses were performed at the ISO 9001-approved main laboratory (Petaluma, Labcon, CA, USA) of the Vincent Group Austria at St. Vincent Hospital in Vienna, Austria.

The central tendency and dispersion of continuous demographic data and serum values (for baseline and follow-up) were described as median and as first and third quartile, respectively. Differences between the study groups were assessed by two-sample t-tests with Welch correction or Mann–Whitney U tests for continuous variables, depending on distributional assumptions, as well as by χ^2^-tests for categorial variables. Longitudinal serum data by study group were visualized by line plots connecting mean values for each time point, and parametric 95% confidence intervals were added. Linear regression models were estimated with serum data as the dependent variable and group and baseline serum data as the explanatory variable for week 2, 4, and 12. In cases of deviations from the normality of residuals, the serum data were log-transformed. This was the case for the iron, ferritin, and transferrin parameters. Distributional assumptions for *t*-tests and linear regressions were checked visually using quantile–quantile plots, which included the corresponding confidence bands. All tests were two-sided and *p*-values less than 0.05 were considered statistically significant. All statistical analyses were performed with the statistical software R version 4.23 (R Development Core Team, 2023, Vienna, Austria).

## 3. Results

Demographic data as well as baseline blood values were comparable between both study groups (Table 1). Hb values were below the reference value in both study groups. Within group 2, the baseline 25(OH)D value was below the reference value and the baseline P1NP value was above the reference value. All other baseline serum values of both study groups were within the therapeutic thresholds (Table 1). At 12 weeks, two patients within group 1 were lost to follow-up. At the final follow up, a total of 18 patients were included in the study (group 1: ferric carboxymaltose, *n* = 8; group 2: iron sucrose, *n* = 10).

Two weeks after drug administration, significant differences in phosphate (*p* < 0.001) and ferritin (*p* < 0.001) levels were observed between both study groups. Hb values were below the reference value in both study groups. Within group 1, the phosphate value was below the reference value and the ferritin value was above the reference value. Within group 2, P1NP and CTX values were above the reference value. All other serum values were within the therapeutic threshold (Table 2, Figure 1).

Four weeks after drug administration, significant differences in phosphate (*p* = 0.043) and ferritin (*p* = 0.009) levels were observed between both study groups. The Hb value was below the reference value within group 2. All other serum values were within the therapeutic threshold (Table 2, Figure 1).

Twelve weeks after drug administration, no differences in serum values were observed between both study groups. Hb values were within the therapeutic threshold in both study groups. The P1NP value was above the reference value within group 1. All other serum values were within the therapeutic thresholds (Table 2, Figure 1).

## 4. Discussion

This study revealed significant differences in serum phosphate levels as well as in serum ferritin levels two and four weeks after intravenous iron substitution therapy when comparing ferric carboxymaltose and iron sucrose treatments. Patients treated with ferric carboxymaltose presented with transient hypophosphatemia as rapidly as two weeks after drug administration, which normalized 12 weeks after intravenous iron substitution therapy. However, P1NP remained upregulated at the final follow-up after treatment with ferric carboxymaltose.

In healthy adults, the serum phosphate level ranges between 0.8 mmol/L and 1.45 mmol/L, with changes across the day in relation to dietary phosphate consumption [5]. Hypophosphatemia can be categorized as mild (0.65 mmol/L to 0.80 mmol/L), intermediate (0.32 mmol/L to 0.65 mmol/L), or severe (1.0 mmol/L to 2.0 mg/dL) [6]. Hypophosphatemia induced by intravenous iron substitution therapy was first described in 1983 [1]. Symptoms of hypophosphatemia range from mild to severe, including muscle weakness, fatigue, myalgia, and osteomalacia [7,8,9,10]. However, osteomalacia is the main prolonged complication of sustained iron-induced hypophosphatemia after multiple doses, which is associated with severe bone pain, nausea, and pseudofractures [3,5].

Under intravenous iron substitution therapy with ferric carboxymaltose, patients presented with hypophosphatemia two weeks after drug administration. This was not observed under iron substitution with iron sucrose. Four weeks after iron substitution therapy, serum phosphate levels were within the therapeutic threshold in both study groups; however, significant differences were still observed when comparing ferric carboxymaltose and iron sucrose. These observations are consistent with those from the literature. A new systematic review by Glaspy et al. showed a higher incidence of hypophosphatemia after treatment with ferric carboxymaltose (up to 92%) compared to after treatment with iron sucrose (up to 40%) [11]. Our understanding of transient hypophosphatemia after intravenous iron substitution therapy comes from studies on autosomal dominant hypophosphatemic rickets, a phosphate wasting disorder [3]. In this disorder, hypophosphatemia has been linked with iron depletion and elevated FGF23 [12]. This FGF23-mediated reduction in serum phosphate levels is generally transient after one single iron infusion [3]. On the other hand, decreased phosphate levels two weeks after treatment with ferric carboxymaltose must be treated with caution, since the previous history of serum phosphate levels remains unknown. Additionally, no data on previous iron substitution therapies or clinical symptoms of hypophosphatemia were available among all study patients. Moreover, hypophosphatemia after ferric carboxymaltose treatment was only observed over two weeks. The clinical impact of transient hypophosphatemia over a time period of two weeks remains unknown. However, prolonged administration may result in long-term complications following hypophosphatemia, such as osteomalacia and bone fractures.

Ferritin represents an indirect reflection of the total body iron stores [13]. Inflammatory syndrome, metabolic syndrome, and alcoholism are mainly responsible for elevated ferritin levels, accounting for over 90% of all cases [13,14]. Chronic hematologic disorders as well as overloads caused by excessive supplementation or infusions can result in hepatic iron overload [14]. Addison et al. emphasized the association between serum ferritin levels and all iron stores [15]. The authors investigated serum ferritin in healthy individuals, individuals with iron insufficiency, and individuals with iron excess. They showed that serum ferritin was raised in the population with an excess of iron and lowered in the population with iron deficiency disorders [16]. Low serum ferritin is an indicator of iron deficiency anemia [17]. Off all the diagnostic values used in the assessment of iron deficiency anemia, serum ferritin was demonstrated to be by far the most conclusive test for diagnosing iron deficiency [18,19]. Limited primary evidence exists from high-quality studies to support specific thresholds. In a study involving 238 healthy women, a ferritin threshold below 15 µg/L accurately predicted the absence of iron stores in the bone marrow, demonstrating a sensitivity of 75% and specificity of 98%. Increasing the threshold to 30 µg/L enhanced the sensitivity to 93%, but reduced the specificity to 75% [20]. Due to the fact that ferritin is an acute-phase protein, the identification of iron deficiency can be obscured by the presence of inflammation [21,22]. In the case of inflammation, raising the ferritin threshold may be considered [23].

In this study, serum ferritin levels were markedly above the reference value two weeks after iron substitution therapy with ferric carboxymaltose. Four weeks after drug administration, significant differences were still observed between both study groups. However, median ferritin levels were within the therapeutic threshold in both study groups. Indeed, elevated ferritin levels after iron substitution therapy with ferric carboxymaltose were only observed for two weeks. These observations must be treated with caution since the patient’s ferritin levels before the baseline measurements were unknown.

P1NP and CTX are both so-called bone turnover markers (BTM), which are generated during bone remodeling. Using serum P1NP, bone formation can be monitored, whereas CTX is a BTM for the bone resorption process [24]. In the case of imbalanced BTMs, increased bone formation or bone resorption can be diagnosed [25]. Nagy et. al reported that increased BTMs are linked to elevated bone turnover, leading to a decrease in bone quality and an increased fracture risk [26]. Serum P1NP and CTX are recommended by the International Osteoporosis Foundation and the International Federation of Clinical Chemistry as reference markers for bone formation as well as for bone resorption. This recommendation is also used for the prediction of fracture risk and to monitor ongoing osteoporosis therapy [27]. However, these markers cannot be used as an independent diagnostic tool; thus, the concomitant measurement of bone mineral density is currently essential [28,29].

In this study, no significant differences in P1NP and CTX were observed between both study groups at any timepoint. The reference value for P1NP is between 15–59 µg/L, and for CTX it is <57 ng/mL. However, patients treated with iron sucrose had increased serum levels, being 68.71 µg/L for P1NP and 0.60 ng/mL for CTX two weeks after iron substitution therapy. Since both values for bone formation and for bone resorption were elevated, this can theoretically imply balanced bone remodeling. Still, as already stated by Nagy et al., the increased bone remodeling rate can result in a decrease in bone quality [26]. However, only four weeks after treatment with iron sucrose, P1NP and CTX values were again within the therapeutic threshold. Interestingly, the serum P1NP value was slightly above the reference value 12 weeks after treatment with ferric carboxymaltose, theoretically resulting in increased bone formation. However, this might be a reaction to the previous transient hypophosphatemia two and four weeks after iron substitution therapy with ferric carboxymaltose. What clinical impact the slightly elevated P1NP level has cannot be estimated. Additionally, in cases of prolonged iron substitution therapy with ferric carboxymaltose, even higher values of P1NP can be expected. Furthermore, at the final follow-up, serum iPTH levels were within the therapeutic thresholds in both study groups. Nevertheless, a trend was observed where there were higher serum iPTH levels within the iron sucrose group compared to the ferric carboxymaltose group. These changes in BTMs that occur as fast as 2 weeks after intravenous iron substitution therapy with remaining differences at final follow-up underline the often neglected impact of iron supplementation on bone homeostasis.

Previous studies demonstrated that intravenous administration of iron carboxymaltose causes hypophosphatemia in up to 75% of cases [30,31]. Furthermore, repeated administration of iron carboxymaltose was linked to serious complications such as muscle weakness, osteomalacia, and bone fractures. These risks are especially elevated for patients with a severe iron deficiency resulting from persistent blood loss [32]. The recently published PHOSPHARE-IBD study examined 97 patients with chronic inflammatory bowel disease and iron deficiency anemia who were treated with intravenous iron carboxymaltose or intravenous iron isomaltoside. The authors showed that there was a significant increase in hypophosphatemia in patients treated with iron carboxymaltose. However, both preparations were equally efficient in correcting iron deficiency anemia. Interestingly, patients who developed hypophosphatemia experienced a prolonged recovery from fatigue symptoms. Additionally, the study demonstrated the metabolic consequences of hypophosphatemia in patients treated with iron carboxymaltose, which led to a decrease in 1,25-dihydroxy-vitamin D and an increase in parathyroid hormone and bone-specific alkaline phosphatase, along with a greater decrease in the anabolic bone marker P1NP [33].

The potential impact of phosphate on bone fractures is still under investigation [34]. Previous cross-sectional research has observed higher serum phosphate levels in patients undergoing hemodialysis who have previously experienced fragility fractures. More precisely, an increase in serum phosphate levels by one milligram per deciliter was linked to an increased risk of 12% for hospitalization due to bone fractures [35]. Campos-Obando et al. reported a correlation between an increased risk for bone fractures and elevated serum phosphate levels in patients with normal kidney function, notably within the standard threshold of serum phosphate levels [35]. Comparable results were reported in large, retrospective observational studies, which found an association between bone fracture risk and serum phosphate levels, even within the physiological serum phosphate range. This correlation appears to be stronger in patients with chronic kidney disease, suggesting that high phosphate levels may explain the increased risk for fractures in this population [36]. Similarly, a retrospective cross-sectional analysis of renal transplant recipients demonstrated that lower serum phosphate levels were associated with an increased risk for bone fractures [37]. Most recently, a follow-up analysis of the PHOSPHARE-IDA study, which was an investigation of serum phosphate levels in patients with iron deficiency anemia, was able to show that an intravenous treatment with ferric carboxymaltose was in fact the only consistent risk factor for the development of hypophosphatemia, which was also related to more severe interactions in bone and mineral metabolism [38].

Vitamin D is crucial for physiological musculoskeletal function as it ensures the absorption of calcium, promotes osteoid tissue mineralization in bones, and preserves muscle function [39]. In cases of low serum vitamin D levels, secondary hyperparathyroidism, bone loss, and muscle weakness can occur [39,40,41,42]. Furthermore, reduced serum 25(OH)D levels are associated with an increased risk of falling and resulting bone fractures [39,43]. On the other hand, meta-analyses demonstrated that vitamin D supplementation may reduce the risk of fractures in older patients [44,45]. Looking at the biochemistry, vitamin D seems to have a function in all stages of bone fracture healing by facilitating calcium mobilization. However, there are contrasting results with regard to the levels of metabolites during the healing process, and the exact mechanism is still not fully understood [46]. In 2017, Sprague et al. reported a high prevalence of hypovitaminosis D in patients with fractures [47].

Interestingly, in our study, serum 25(OH)D levels remained within the therapeutic threshold within both study groups throughout the whole study period. However, 25(OH)D values continually declined in both study groups until the last follow-up.

There are several limitations to this study. Due to the small sample sizes, the results must be interpreted with caution. However, even with this small sample size, such highly significant differences in serum phosphate levels could be detected by comparing both drugs. Even if a longitudinal assessment of serum markers, including baseline values, was performed, no information was available regarding the previous serum values of all patients, previous iron substitution therapies, and clinical symptoms of hypophosphatemia or elevated ferritin levels. Furthermore, no data on FGF23 levels were available. This may be of great interest for future studies, since FGF23 levels are expected to drastically change after intravenous iron substitution therapy, along with the resulting changes in serum phosphate levels and transient hypophosphatemia.

## 5. Conclusions

In conclusion, the intravenous administration of ferric carboxymaltose and iron sucrose led to significant changes in serum phosphate and ferritin levels, with transient hypophosphatemia occurring as rapidly as two weeks after drug application. Even four weeks after treatment, these changes in serum phosphate and ferritin levels were still observed. However, twelve weeks after drug administration, no differences were observed in the serum phosphate and ferritin levels anymore. All serum values were within the therapeutic threshold. Anemia and iron deficiency were successfully treated with both ferric carboxymaltose and iron sucrose.

## Figures and Tables

**Figure 1 nutrients-15-02693-f001:**
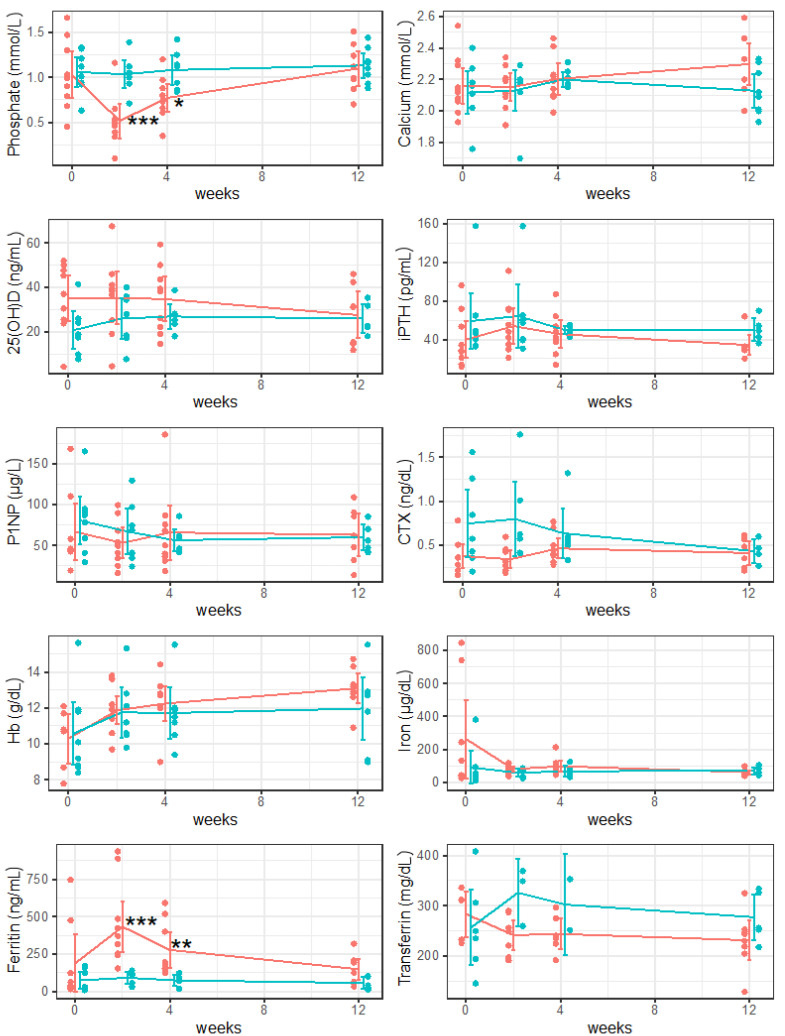
Longitudinal evaluation of serum values 2, 4, and 12 weeks after drug administration; group 1 (ferric carboxymaltose) presented in orange and group 2 (iron sucrose) presented in turquoise. 25(OH)D, 25-hydroxyvitamin D; CTX, beta-CrossLaps; Hb, hemoglobin; iPTH, intact parathyroid hormone; P1NP, procollagen type 1 amino-terminal propeptide. Data are presented as mean and 95% confidence intervals. * *p* < 0.05; ** *p* < 0.01; *** *p* < 0.001.

**Table 1 nutrients-15-02693-t001:** Demographic data and baseline blood values, group 1 (ferric carboxymaltose) and group 2 (iron sucrose).

	Ferric Carboxymaltose	Iron Sucrose	*p*-Value
Age	48 [34;57]	63 [46;72]	0.558
Gender (female/male)	5/3	6/4	0.999
Height (cm)	167 [163;172]	167 [163;181]	0.552
Weight (kg)	67 [49;74]	72 [38;87]	0.852
BMI (kg/m^2^)	21.7 [18.7;26.8]	23.2 [14.5;29.9]	0.883
Phosphate (0.60–1.55 mmol/L)	0.99 [0.79;1.30]	1.07 [0.94;1.24]	0.848
Calcium (2.10–2.58 mmol/L)	2.13 [2.07;2.26]	2.14 [2.02;2.20]	0.609
25(OH)D (20–70 ng/mL)	37.00 [25.50;47.60]	18.80 [13.60;24.95]	0.051
iPTH (15–65 pg/mL)	28.40 [20.90;53.50]	47.00 [38.60;53.10]	0.200
P1NP (15–59 µg/L)	45.76 [43.68;71.00]	82.61 [54.39;91.45]	0.505
CTX (<0.57 ng/mL)	0.36 [0.26;0.40]	0.57 [0.39;1.05]	0.152
Hb (12–16 g/dL)	10.80 [9.20;11.50]	9.65 [8.78;11.80]	0.817
Iron (37–145 µg/dL)	88.50 [38.30;369.00]	43.00 [28.50;74.00]	0.336
Ferritin (5–204 ng/mL)	50.00 [28.25;211.25]	30.00 [10.50;141.50]	0.417
Transferrin (200–360 mg/dL)	311.00 [233.00;312.00]	242.50 [204.25;292.00]	0.553

25(OH)D, 25-hydroxyvitamin D; BMI, body mass index; CTX, beta-CrossLaps; Hb, hemoglobin; iPTH, intact parathyroid hormone; P1NP, procollagen type 1 amino-terminal propeptide. Data are presented as median and interquartile ranges Q1 and Q3.

**Table 2 nutrients-15-02693-t002:** Longitudinal evaluation of serum values 2, 4, and 12 weeks after drug administration for group 1 (ferric carboxymaltose) and group 2 (iron sucrose).

	Ferric Carboxymaltose	Iron Sucrose	*p*-Value
**Phosphate (0.60–1.55 mmol/L)**			
2 weeks	0.50 [0.44;0.57]	1.04 [0.97;1.09]	**<0.001**
4 weeks	0.74 [0.66;0.88]	1.12 [0.90;1.12]	**0.043**
12 weeks	1.06 [0.95;1.27]	1.13 [1.01;1.22]	0.725
**Calcium (2.10–2.58 mmol/L)**			
2 weeks	2.19 [2.09;2.21]	2.18 [2.17;2.19]	0.908
4 weeks	2.19 [2.10;2.23]	2.17 [2.16;2.23]	0.957
12 weeks	2.33 [2.19; 2.36]	2.11 [2;20.26]	0.1
**25(OH)D (20–70 ng/mL)**			
2 weeks	38.50 [25.20;41.00]	28.00 [17.80;35.15]	0.987
4 weeks	38.20 [22.10;43.50]	26.55 [23.95;27.95]	0.942
12 weeks	31.10 [15.00;36.90]	22.70 [22.10;31.70]	0.57
**iPTH (15–65 pg/mL)**			
2 weeks	48.00 [35.20;70.80]	57.40 [39.75;62.75]	0.516
4 weeks	42.10 [37.20;53.40]	50.85 [45.40;53.53]	0.578
12 weeks	31.40 [29.40;32.30]	49.00 [42.60;54.40]	0.075
**P1NP (15–59 µg/L)**			
2 weeks	49.55 [34.55;68.12]	68.71 [37.93;85.80]	0.859
4 weeks	50.20 [36.46;75.81]	53.12 [44.20;61.02]	0.218
12 weeks	61.93 [40.17;88.00]	56.16 [47.46;70.00]	0.618
**CTX (<0.57 ng/mL)**			
2 weeks	0.29 [0.25;0.43]	0.60 [0.45;0.91]	0.199
4 weeks	0.40 [0.38;0.50]	0.54 [0.50;0.58]	0.717
12 weeks	0.41 [0.27;0.55]	0.43 [0.36;0.60]	0.824
**Hb (12–16 g/dL)**			
2 weeks	11.90 [11.45;12.12]	11.20 [10.55;12.45]	0.85
4 weeks	12.10 [12.00;12.80]	11.50 [10.90;12.00]	0.13
12 weeks	13.10 [12.80;13.60]	12.70 [10.50;12.90]	0.073
**Iron (37–145 µg/dL)**			
2 weeks	80.00 [59.80;101.00]	68.00 [36.00;78.00]	0.342
4 weeks	89.00 [66.00;115.00]	60.00 [44.00;80.00]	0.273
12 weeks	61.00 [49.00;66.50]	60.00 [58.00;85.00]	0.770
**Ferritin (5–204 ng/mL)**			
2 weeks	344.50 [252.50;471.00]	111.00 [55.00;119.00]	**<0.001**
4 weeks	191.00 [154.00;401.00]	71.00 [70.00;89.00]	**0.009**
12 weeks	159.00 [61.00;200.25]	41.00 [25.00;100.00]	0.109
**Transferrin (200–360 mg/dL)**			
2 weeks	249.00 [209.50;271.50]	349.00 [304.50;359.00]	0.326
4 weeks	238.00 [227.50;266.50]	302.50 [277.25;327.75]	0.831
12 weeks	237.00 [215.50;249.25]	256.00 [253.00;326.00]	0.324

25(OH)D, 25-hydroxyvitamin D; CTX, beta-CrossLaps; Hb, hemoglobin; iPTH, intact parathyroid hormone; P1NP, procollagen type 1 amino-terminal propeptide. Data are presented as median and interquartile ranges Q1 and Q3. Significant values are marked in bold.

## Data Availability

The datasets used and analyzed in this study are available from the corresponding author upon request.

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
