# Peer review of "Impact of Intravenous Iron Substitution on Serum Phosphate Levels and Bone Turnover Markers—An Open-Label Pilot Study"

_nutrients, 2023, doi:10.3390/nu15122693_

Round 1

Reviewer 1 Report

In this study, Struppe et al investigated bone markers and phosphate serum levels after intravenous iron treatment. This study is of undoubted scientific interest.

However I found serious flaws in methodology with very high risk of bias. The presentation of the results is not optimal. The discussion must be improved also.

Please find below my comments.

MAJOR

1.       One of the main issue in this study is the lack of precision in the inclusion criteria. The authors stated they included patients

·         With IBD with chronic occult bleeding. How bleeding is assessed? Fecal test?

·         With “another underlying disease”: heart failure? Malabsorption disease?

As such, how did the authors proceeded to maintain the comparability between the two arms? Did they use stratification? With so few details provided by the authors, the lack of differences in baseline characteristics makes me feel very uncomfortable about the risk of bias selection

2.       The second main issue is the lack of sample size calculation: the authors stated no clinical or biological difference expected between the two groups. This fact is very surprising as some studies are available with variance of phosphate serum level after iron treatment.

In any case the authors should discuss the statistical power in the light of the very limited sample size (with 2 lost-of-follow-up in one group!).

3.       The third issue is the lack of precision about concomitant treatment that may have influenced bone markers levels during the study period. How did the authors proceeded to ensure that no modifications about treatment and nutritional behaviours occurred between D0, W2, W4 and W12? I found nothing about this important issue in the manuscript.

4.       I recommend the authors to compare their own results to the literature.. In their systematic literature review, Glaspy et al highlighted a highly heterogeneous prevalence ranging from 0 to 40%

MINOR

General remarks:

In most published papers, sucroferric oxyhydroxide is named iron sucrose. For a non-expert reader, I suggest the authors to use “iron sucrose” rather than sucroferric oxyhydroxide, which name refers most often to the oral tablet used as a phosphate chelator for patients with chronic kidney disease. Otherwise, please justify the use of sucroferric oxyhydroxide rather than iron sucrose.

Title:

I suggest the authors to add the open label design in the title.

Abstract:

The authors should improve the abstract. I first read the abstract without reading the manuscript in extenso and failed to clearly identify the studied population (lines 17-18):

-          The authors should clarify immediately that the patients were followed for IBD

Clearly state the open-label procedure used in the abstract.

Line 17: How the authors calculated the number of patients to include? If no preliminary hypothesis of differences was used to calculate the sample size, the authors should justify it.

Line 26: Please specify the “therapeutic range” of phosphate and ferritin levels, and eventually justify the choice of these values.

Line 29: it would be of interest to specify in which group ferritin and phosphate level were higher.

Lines 33-38 are redundant.

 Methods:

1.       Could the authors explain why they chose iron sucrose and carboxymaltose rather than other iron formulations iron isomaltoside or iron dextran or ferumoxytol?

2.       In most countries the maximum single dose of ferric carboxymaltose in a single dose is 750 – 1000 mg: as such I am surprise to see a use of 1500 mg in a single dose. Moreover in my country, the dosage of FC is recommended to be chosen according to ferritin, body weight and haemoglobin level. Could the authors provide a brief explanation?

3.       Similarly, for SO, in my country, the recommended dosage is 100 to 200 mg per injection, 1 to 3 times per week, with a minimum interval of 48 hours between each infusion because the repeated infusion increase the risk of complications. Can the authors comment the daily infusions?

4.       The authors stated they collected blood samples 2-4-12 weeks after iron treatment: can they be more accurate? Is it X weeks after the last infusion of iron supplement (as SO is given in 3 infusions

5.       Can the authors explain why they use only visual comparison to confirm the assumption of homoscedasticity and not the Fisher Snedecor test? Please confirm that the assumptions of t test were met for each comparison and if not, please use the appropriate Mann-Whitney test (the very low sample size is at high risk…).

Results

1.       The level of baseline ferritin (50 and 30 µg/L) is quite high to initiate an iron supplementation. Have the authors assessed the C reactive protein at baseline?

It is therefore crucial to state the indication for iron supplementation, to clearly identify the included population and to let the readers evaluate the risk of confounding bias in the study of bone metabolism markers.

2.       Table 2 and figure 1 are redundant. Please choose table or figure and place the other in supplements. I suggest to present only significant variations if the authors chose the figure.

3.       The authors present all their results with Inter quartile range: if so, I assume the variables do not follow a normal distribution… so the use of t test is questionable.

Discussion:

1.       Please turn the discussion in a more concise way to improve the readability, some general statements might be deleted.

·         The first section (L161-184) could be summarized in 3 to 4 lines.

2.       A word seems to be missing in the sentence Lines 176-178 “However, it is important to emphasize, that the risk and severity of hypophosphatemia is dependent on the severity, chronicity, and duration of recovery”

3.       In the second section (lines 185-198) the authors should compare the differences between their results and those published in the literature in term of hypophosphatemia prevalence with iron sucrose (as the authors did in the section lines 265-279. I suggest to merge these two sections.

4.       I do not understand the sentence “pre-treatment phosphate levels were unknown”, I thought the authors had measured the serum phosphate on the day of the 1st iron injection?

5.       The use of the word “range” is weird to argue about the “normal” values for ferritin.

The use of thresholds vary significantly in the literature according to the existence of an underlying disease (CKD, IBD, cardiac chronic heart failure), of age and sex. Thus, some authors use 20 µg/L for premenopausal women because it is associated with an increased risk of anaemia but other authors use higher thresholds because fatigue and other symptoms are significantly more frequent under 50 µg/L. Another way to define ferritin thresholds is the therapeutic efficacy of iron supplementation: as such in CHF, 100 µg/L is used because iron treatment has been associated to an improved functional status. I think this part of the discussion should be improved.

Thus the reference 18 is not the most recent one about the ferritin “normal values”.

6.       Lines 214-231: It is not very relevant to write a section on iron overload, especially since it is a complex subject and some of the authors' statements are, at best, approximate and, at worst, highly debatable. Treating patients with iron supplement will never induce iron overload if the indication for treatment is correct (such as chronic haemorrhage !). So please improve significantly the argument or delete this section.

7.       What do the authors mean when they state that P1NP and CTX increase “beyond the references”: is it statistically significant? Clinically relevant? Be more precise please.

Author Response

Responses to the Reviewers

Manuscript nutrients-2421876: Impact of intravenous iron substitution on serum phosphate levels and bone turnover markers – An open-label pilot study

Dear Ladies and Gentlemen,

We are very grateful to the Editor and the Reviewers for their expertise and have addressed all comments and suggestions brought forward. Please find below the details of our revision.

The responses to the Reviewer comments are marked as Response in this document. Changes in the revised manuscript and in this document are marked in turquoise. Deleted text is marked as strikethrough in the revised manuscript and in this document. Changes not related to the Reviewers’ comments are only corrected in the revised manuscript. Since no bibliography was added to the Responses to the Reviewers, all citations were removed from this document. In the manuscript, citations and bibliography were updated. Page and line numbering is referred to the marked and revised manuscript. Please notice that the title of the manuscript was slightly adapted, as suggested by one Reviewer.

Reviewer #1

Comments and Suggestions for Authors:

In this study, Struppe et al investigated bone markers and phosphate serum levels after intravenous iron treatment. This study is of undoubted scientific interest. However, I found serious flaws in methodology with very high risk of bias. The presentation of the results is not optimal. The discussion must be improved also. Please find below my comments.

Major.

Response: We thank the Reviewer for his great interest in our study and his important remarks. As suggested, all comments of the Reviewer were addressed in our revised manuscript.

  1. One of the main issues in this study is the lack of precision in the inclusion criteria. The authors stated they included patients

- With IBD with chronic occult bleeding. How bleeding is assessed? Fecal test?

- With “another underlying disease”: heart failure? Malabsorption disease?

As such, how did the authors proceeded to maintain the comparability between the two arms? Did they use stratification? With so few details provided by the authors, the lack of differences in baseline characteristics makes me feel very uncomfortable about the risk of bias selection.

Response: We thank the Reviewer for this very important comment. Patients with IBD with occult blood loss were identified by a fecal test. In our study, patients with “another underlying disease” presented an iron deficiency anemia. In total, we thereby included patients with IBD and patients with iron deficiency anemia. We changed the Abstract as well as the Methods section of the main manuscript to be more precise concerning patient inclusion criteria. The paragraphs reads as follows:

“Inclusion criteria were an underlying chronic inflammatory bowel disease (Crohn’s Disease, ulcerative colitis) with occult blood determined by a fecal test or another underlying disease requiring intravenous iron substitution such as iron deficiency anemia. The diagnosis was based on serum levels presenting with microcytic hypochromic anemia or fecal test.” (page 2, lines 86-90).

  1. The second main issue is the lack of sample size calculation: the authors stated no clinical or biological difference expected between the two groups. This fact is very surprising as some studies are available with variance of phosphate serum level after iron treatment. In any case the authors should discuss the statistical power in the light of the very limited sample size (with 2 lost-of-follow-up in one group!).

Response: We thank the Reviewer for this important comment. Definitely, the small sample size, especially with 2 patients lost to follow-up in one group is low. However, we observed significant differences of phosphate as well as ferritin levels between both study groups even with these small sample sizes and in a short follow-up period of only 2 weeks. Furthermore, we want to refer to another Reviewer’s comment, as “examined pts but final results can justify it”. Nevertheless, as correctly identified by the Reviewer due to the small sample sizes, the results must be interpreted with caution. Also, we want to clarify that this investigation was planned as a pilot study, thereby no a priori sample size calculation is necessary. This methodical issue was added in the title as well as the main manuscript.

  1. The third issue is the lack of precision about concomitant treatment that may have influenced bone markers levels during the study period. How did the authors proceeded to ensure that no modifications about treatment and nutritional behaviors occurred between D0, W2, W4 and W12? I found nothing about this important issue in the manuscript.

Response: We thank the Reviewer for this very important comment. Patients received intravenous iron substitution solely in the hospital according to the dose protocols stated in the manuscript as well as pain medication as needed. As clarified in the ethical approval as well as in the informed consent form, patients were instructed not to receive any other treatments during the study period besides their ongoing long-term medication, as this could have affected the study outcomes. If any further treatments would have been necessary, patients are obliged to report this to the study investigators. None of the patients reported any further dietetical change or adaptation of medication during the whole study period. An explanation was added to the Materials and Methods section of the main manuscript and reads as follows: “Patients were further instructed not to receive any other treatments during the study period besides their ongoing long-term medication, as this could have affected the study outcome. Also, patients should not receive any dietetical changes or adaptations of medication.“ (pages 2, lines 93-96).

  1. I recommend the authors to compare their own results to the literature. In their systematic literature review, Glaspy et al highlighted a highly heterogeneous prevalence ranging from 0 to 40%.

Response: We thank the Reviewer for this comment. This important systematic review was added. Our findings were discussed and compared to this recently published systematic review.

„Under intravenous iron substitution therapy with ferric carboxymaltose, patients presented with hypophosphatemia two weeks after drug administration. This was not observed under iron substitution with iron sucrose sucroferric oxyhydroxide. Four weeks after iron substitution therapy, serum phosphate levels were within the therapeutic threshold range in both study groups, however, significant differences were still observed comparing ferric carboxymaltose and iron sucrose sucroferric oxyhydroxide. These observations are consistent with the literature. A new systematic review by Glaspy et al. showed a higher incidence of hypophosphatemia after treatment with ferric carboxymaltose (up to 92%) compared to treatment with iron sucrose (up to 40%) [12].“ (page 7, lines 210-213).

Minor:

General remarks:

In most published papers, sucroferric oxyhydroxide is named iron sucrose. For a non-expert reader, I suggest the authors to use “iron sucrose” rather than sucroferric oxyhydroxide, which name refers most often to the oral tablet used as a phosphate chelator for patients with chronic kidney disease. Otherwise, please justify the use of sucroferric oxyhydroxide rather than iron sucrose.

Response: We thank the Reviewer for this thoughtful comment. The term “sucroferric oxyhydroxide” was changed throughout the whole manuscript to “iron sucrose” as suggested.

Title: I suggest the authors to add the open label design in the title.

Response: As suggested by the Reviewer, the title was changed and reads as follows:

“Impact of intravenous iron substitution on serum phosphate levels and bone turnover markers – An open-label pilot study” (page 1, lines 1-3).

Abstract: The authors should improve the abstract. I first read the abstract without reading the manuscript in extenso and failed to clearly identify the studied population (lines 17-18):

- The authors should clarify immediately that the patients were followed for IBD

Clearly state the open-label procedure used in the abstract.

Response: The abstract was modified accordingly to the Reviewer’s remark and reads as follows:

A In this open-label pilot study, a total of 20 patients with inflammatory bowel diseases or iron deficiency anemia were randomly assigned to one of both study groups (group 1: ferric carboxymaltose, n=10; group 2: iron sucrose sucroferric oxyhydroxide, n=10).” (page 1, lines 17-20).

Line 17: How the authors calculated the number of patients to include? If no preliminary hypothesis of differences was used to calculate the sample size, the authors should justify it.

Response: We thank the Reviewer for this thoughtful comment. This study was planed as a pilot study, so no a priori sample size calculation was performed. This was added in the Title, the Abstract as well as the Methods.

Line 26: Please specify the “therapeutic range” of phosphate and ferritin levels, and eventually justify the choice of these values.

Response: We thank the Reviewer for this comment. The therapeutic ranges according to the reference values of the laboratory for phosphate, ferritin as well as P1NP and CTX were added to the Abstract:

“Phosphate levels (0.8-1.45mmol/L) were below the therapeutic threshold range and ferritin levels (10-200 ng/mL for women and 30-300ng/mL for men) were above the therapeutic threshold range in group 1. P1NP (15-59µg/l) and CTX (<0.57ng/ml) were above the therapeutic threshold range in group 2. Four weeks after drug administration, significant differences were still observed between both study groups for phosphate (p=0.043) and ferritin (p=0.032 p=0.0009).” (page 1, lines 27-32).

Line 29: It would be of interest to specify in which group ferritin and phosphate level were higher.

Response: We thank the Reviewer for this valid comment. The Abstract was modified as suggested and reads as follows:

“In group 1, serum phosphate levels were lower and serum ferritin levels were higher compared to group 2.” (page 1, lines 32-33).

Lines 33-38 are redundant.

Response: We thank the Reviewer for this comment. As suggested, these lines were deleted in the revised manuscript.

Methods:

  1. Could the authors explain why they chose iron sucrose and carboxymaltose rather than other iron formulations iron isomaltoside or iron dextran or ferumoxytol?

Response: Ferric carboxymaltose and iron sucrose are the two available intravenous iron substitutes used in the St. Vincent Hospital Vienna. Both drugs are widely tested and commonly used in clinical routine. Furthermore, ferric carboxymaltose and iron sucrose have comparable efficacy in correcting iron deficiency, but differ in their infusion reactions and the daily dose that can be applied.

  1. In most countries the maximum single dose of ferric carboxymaltose in a single dose is 750 – 1000 mg: as such I am surprise to see a use of 1500 mg in a single dose. Moreover, in my country, the dosage of FC is recommended to be chosen according to ferritin, body weight and haemoglobin level. Could the authors provide a brief explanation?
  2. Similarly, for SO, in my country, the recommended dosage is 100 to 200 mg per injection, 1 to 3 times per week, with a minimum interval of 48 hours between each infusion because the repeated infusion increase the risk of complications. Can the authors comment the daily infusions?

Response: We thank the Reviewer for this very important comment. The application of both drugs was performed according to the Austria Codex, the Austrian national registry on use of drugs and medication. Please note, that the dose in the manuscript are the maximum doses as suggested by the pharmaceutical companies. We reviewed all our drug administration protocols and found similar doses as mentioned by the Reviewer. Also, no complications or adverse events were observed after drug administration. This was adapted in the manuscript:

“Group 1 received one single-dose of 1500mg 1000mg ferric carboxymaltose (Ferinject®, CSL Vifor, St. Gallen, Swizerland) intravenously. Group 2 was treated with 200mg iron sucrose sucroferric oxyhydroxide (FerMed®, Medice Arzneimittel GmbH, Hallein, Austria) intravenously once daily for three consecutive days.” (page 2, lines 96-99).

  1. The authors stated they collected blood samples 2-4-12 weeks after iron treatment: can they be more accurate? Is it X weeks after the last infusion of iron supplement (as SO is given in 3 infusions)?

Response: We thank the Reviewer for this comment. More precisely, blood samples were collected 2, 4, and 12 weeks after the last intravenous iron administration in both study groups. This was added to the manuscript:

“Blood samples were collected initially before iron substitution as well as 2, 4, and 12 weeks after the last intravenous iron administration later.” (page 3, lines 101, 102).

  1. Can the authors explain why they use only visual comparison to confirm the assumption of homoscedasticity and not the Fisher Snedecor test? Please confirm that the assumptions of t test were met for each comparison and if not, please use the appropriate Mann-Whitney test (the very low sample size is at high risk…).

Response: Baseline (Table 1): We applied t-tests with Welch approximation for unequal variances/heteroscedasticity. We did not apply a prior test on homoscedasticity following current recommendations (Zimmerman, DW. A note on preliminary tests of equality of variances. British Journal of Mathematical and Statistical Psychology. 2004, 57(1), 173-181. doi: 10.1348/000711004849222). Normality was assessed visually because of low sample size. QQ-Plots show deviations of normality for the following parameters: "iPTH", "P1NP", "CTX", "iron", "Ferritin". Therefore, Mann-Whitney tests were carried out for these parameters.

Longitudinal comparisons at week 2, 4, 12 (Table 2): The applied model was not correctly specified in the method section: a linear regression model with the respective parameter as dependent variable at week 2, 4, 12 and group plus baseline value as explanatory variable was estimated. For parameters iron, ferritin, and transferritin, values were log-transformed in order to meet assumptions.

Statistical methods as well as interpretation were accordingly changed throughout the manuscript (Abstract, Material and Methods, Results, Discussion). Please notice that Figure 1 was changed according to the new statistical calculations and results.

Results

  1. The level of baseline ferritin (50 and 30 µg/L) is quite high to initiate an iron supplementation. Have the authors assessed the C reactive protein at baseline?

Response: We thank the author for this very important comment. C-reactive protein was not addressed in all patients. Due to missing values and the small sample, we decide not to include these values in the manuscript.

It is therefore crucial to state the indication for iron supplementation, to clearly identify the included population and to let the readers evaluate the risk of confounding bias in the study of bone metabolism markers.

Response: We thank the Reviewer for this comment. Iron deficiency was diagnosed by means of decreased serum ferritin and transferrin as well as a complete blood count indicating anemia with microcytic, hypochromic red blood cells. Deficiency or anemia symptoms limit the quality of life of affected patients and should therefore be compensated. For the detection of iron deficiency, the inflammatory activity of the chronic inflammatory bowel disease was taken into account. This is clearly stated in the Methods section:

“Inclusion criteria were an underlying chronic inflammatory bowel disease (Crohn’s Disease, ulcerative colitis) with occult blood determined by a fecal test or another underlying disease requiring intravenous iron substitution such as iron deficiency anemia. The diagnosis was based on serum levels presenting with microcytic hypochromic anemia or fecal test. Patients were excluded in case of an underlying oncological disease, an ongoing oral iron substitution therapy, or an acute bleeding anemia. Furthermore, postoperative patients as well as patients requiring intensive care were excluded from the study. Patients were further instructed not to receive any other treatments during the study period besides their ongoing long-term medication, as this could have affected the study outcome. Also, patients should not receive any dietetical changes or adaptations of medication.” (page 2, lines 86-96).

  1. Table 2 and figure 1 are redundant. Please choose table or figure and place the other in supplements. I suggest to present only significant variations if the authors chose the figure.

Response: We thank the Reviewer for this valid comment. However, we want to stress out the fact, that longitudinal evaluation of serum levels can easily be observed by Figure 1. Nevertheless, Table 2 is also highly important for the readers to see exact data. We therefore want to give the decision to the Editor whether to choose Figure 1, Table 2, or both.

  1. The authors present all their results with Inter quartile range: if so, I assume the variables do not follow a normal distribution… so the use of t test is questionable.

Response: We thank the Reviewer for this important comment. Statistical methods are extensively described in the Methods section and changed as previously suggested by the Reviewer:

“Central tendency and dispersion of continuous demographic data and serum values (for baseline and follow-up) were described by median and first and third quartile, respectively. Differences between study groups were assessed by two-sample t-tests with Welch correction or Mann-Whitney-U tests for continuous variables depending on distributional assumptions and by χ2-tests for categorial variables. Longitudinal serum data by study group were visualized by line plots connecting mean values for each time point and parametric 95% confidence intervals were added. Linear regression models were estimated with serum data as dependent variable and group and baseline serum data as explanatory variable for week 2, 4, and 12, respectively. In case of deviations from normality of residuals, serum data was log-transformed. This was the case for parameters iron, ferritin, and transferrin, respectively. Distributional assumptions for t-tests and linear regressions were checked visually by quantile-quantile plots including corresponding confidence bands. All tests were two-sided and p-values less than 0.05 were considered statistically significant. All statistical analyses were performed with the statistical software R version 4.02 4.23 (R Development Core Team, 2018 2023). (page 3, lines 107-121).

Discussion:

  1. Please turn the discussion in a more concise way to improve the readability, some general statements might be deleted.

The first section (L161-184) could be summarized in 3 to 4 lines.

Response: We thank the Reviewer for this comment. As suggested, we changed the first section about serum phosphate and reads as follows:

“In healthy adults, serum phosphate ranges between 0.8mmol/l and 1.45mmol/l with changes across the day in relation to dietary phosphate consumption. Hypophosphatemia can be categorized as mild (0.65 mmol/l to 0.80 mmol/l), intermediate (0.32mmol/l to 0.65 mmol/l), or severe (1.0mmol/l to 2.0 mg/dl). Different grades of hypophosphatemia may also occur in course of chronic diseases, most commonly due to renal phosphate loss or reduced intestinal absorption. There are many etiologies for chronic hypophosphatemia which involve both acquired and genetic dysfunctions. Hypophosphatemia induced by intravenous iron substitution therapy was first described in 1983. Ever since, hypophosphatemia is a well-documented side effect following intravenous iron substitution. The transient hypophosphatemia is due to renal phosphate loss as a result of 25(OH)D resistance, secondary hyperparathyroidism, and iron induced stabilization of FGF23. The proposed underlying cause of this transient hypophosphatemia, after intravenous iron substitution comes from studies of autosomal dominant hypoposphatemic rickets, a phosphate wasting disorder. In these, hypophosphatemia was linked with iron depletion and elevated FGF23. FGF23-mediated reduction in serum phosphate levels is generally transient after one single iron infusion. However, it is important to emphasize, that the risk and severity of hypophosphatemia is dependent on the severity, chronicity, and duration of recovery. Mild to intermediate hypophosphatemia presents mostly asymptomatic. Acute as well as severe hypophosphatemia may result in numerous symptoms. Symptoms of hypophosphatemia range from mild to severe, including such as muscle weakness, fatigue, myalgia, and osteomalacia. paresthesia, and occasionally even heart failure, diaphragmatic weakness, encephalopathy, or hemolytic anemia. However, osteomalacia is the main prolonged complication of sustained iron-induced hypophosphatemia after multiple doses, which is associated with severe bone pain, nausea, and pseudofractures.” (page 7, lines 180-204).

  1. A word seems to be missing in the sentence Lines 176-178 “However, it is important to emphasize, that the risk and severity of hypophosphatemia is dependent on the severity, chronicity, and duration of recovery”.

Response: We thank the Reviewer for this comment. As stated in the previous comment, this paragraph was modified and this sentence was deleted.

  1. In the second section (lines 185-198) the authors should compare the differences between their results and those published in the literature in term of hypophosphatemia prevalence with iron sucrose (as the authors did in the section lines 265-279. I suggest to merge these two sections.

Response: We thank the Reviewer for this thoughtful comment. We modified this paragraph and additionally inserted a new reference as previously suggested by the Reviewer:

“Under intravenous iron substitution therapy with ferric carboxymaltose, patients presented with hypophosphatemia two after drug administration. This was not observed under iron substitution with iron sucrose sucroferric oxyhydroxide. Four weeks after iron substitution therapy, serum phosphate levels were within the therapeutic threshold range in both study groups, however, significant differences were still observed comparing ferric carboxymaltose and iron sucrose sucroferric oxyhydroxide. These observations are consistent with the literature. A new systematic review by Glaspy et al. showed a higher incidence of hypophosphatemia after treatment with ferric carboxymaltose (up to 92%) compared to treatment with iron sucrose (up to 40%) [12]. The proposed underlying cause of this transient hypophosphatemia, after intravenous iron substitution comes from studies of autosomal dominant hypoposphatemic rickets, a phosphate wasting disorder. In these, hypophosphatemia was linked with iron depletion and elevated FGF23. FGF23-mediated reduction in serum phosphate levels is generally transient after one single iron infusion. On the other hand, decreased phosphate levels two weeks after treatment with ferric carboxymaltose must be treated with caution, since a previous history of serum pre-treatment phosphate levels, remains were unknown. Also, no data on previous iron substitution therapies or clinical symptoms of hypophosphatemia were available among all study patients. Moreover, hypophosphatemia after ferric carboxymaltose treatment was only observed over two weeks. The clinical impact of a transient hypophosphatemia over a time-period of two weeks remains unknown. However, prolonged administration may result in long-term complications following hypophosphatemia such as osteomalacia and bone fractures.” (page 7, lines 205-226).

  1. I do not understand the sentence “pre-treatment phosphate levels were unknown”, I thought the authors had measured the serum phosphate on the day of the 1st iron injection?

Response: We thank the Reviewer for this comment. We measured the serum phosphate levels on the day of the 1st intravenous iron substitution to have a baseline level. However, a previous history of phosphate serum levels was unknown. This was clarified and reads as follows:

“On the other hand, decreased phosphate levels two weeks after treatment with ferric carboxymaltose must be treated with caution, since a previous history of serum pre-treatment phosphate levels were remains unknown.” (page 7, lines 218-220).

  1. The use of the word “range” is weird to argue about the “normal” values for ferritin.

The use of thresholds vary significantly in the literature according to the existence of an underlying disease (CKD, IBD, cardiac chronic heart failure), of age and sex. Thus, some authors use 20 µg/L for premenopausal women because it is associated with an increased risk of anemia but other authors use higher thresholds because fatigue and other symptoms are significantly more frequent under 50 µg/L. Another way to define ferritin thresholds is the therapeutic efficacy of iron supplementation: as such in CHF, 100 µg/L is used because iron treatment has been associated to an improved functional status. I think this part of the discussion should be improved. Thus the reference 18 is not the most recent one about the ferritin “normal values”.

Response: We thank the Reviewer for this very important comment. As the term “threshold” is correct, we changed “therapeutic range” to “therapeutic threshold” throughout the whole manuscript. We improved this part of the discussion and leads as follows:

Reference levels are quite variable, but values of 30ng/ml to 300ng/ml are within the therapeutic range for men and 10ng/ml to 200ng/ml for women respectively. Off all the diagnostic values used in the assessment of iron deficiency anemia, serum ferritin was demonstrated to be by far the most conclusive test for diagnosing iron deficiency. Limited primary evidence exists from high-quality studies to support specific thresholds. In a study involving 238 healthy women, a ferritin threshold below 15 µg/L accurately predicted the absence of iron stores in the bone marrow, demonstrating a sensitivity of 75% and specificity of 98%. Increasing the threshold to 30 µg/L enhanced sensitivity to 93% but reduced specificity to 75%. Due to the fact that ferritin is an acute phase protein, the identification of iron deficiency can be veiled by the presence of inflammation. In case of inflammation, raising the ferritin threshold may be considered.” (page 8, lines 237-248).

  1. Lines 214-231: It is not very relevant to write a section on iron overload, especially since it is a complex subject and some of the authors' statements are, at best, approximate and, at worst, highly debatable. Treating patients with iron supplement will never induce iron overload if the indication for treatment is correct (such as chronic haemorrhage !). So please improve significantly the argument or delete this section.

Response: We thank the Reviewer for this valid comment. This section was modified and reads as follows:

On the other hand, ferritin can also be used for identification an iron overload. There is no physiological mechanism for the excretion of an iron overload. Therefore, iron excess is mostly observed in cases of abnormal iron absorption or increased iron application. Iron excess accumulates in the heart and liver and are causing chronic damage of these tissues due to the accumulation of free radicals. Over years, this tissue damage can lead to ongoing liver and heart failure, resulting in substantial morbidity as well as early mortality. Additional symptoms triggered by excess iron include skin changes, endocrine dysfunction, and arthropathy. In this study, serum ferritin levels were markedly above the reference value two weeks after iron substitution therapy with ferric carboxymaltose. Four weeks after drug administration, significant differences were still observed between both study groups. However, median ferritin levels were within the therapeutic threshold range in both study groups. Indeed, elevated ferritin levels after iron substitution therapy with ferric carboxymaltose were only observed for two weeks. However, prolonged administration of ferric carboxymaltose may result in an accumulation of iron which could be observed in upregulated ferritin levels. This may eventually lead to long-term complications as mentioned above, namely concerning the heart and/or the liver. Again, these These observations must be treated with caution since the patient’s ferritin levels before baseline measurements were unknown (page 8, lines 248-266).

  1. What do the authors mean when they state that P1NP and CTX increase “beyond the references”: is it statistically significant? Clinically relevant? Be more precise please.

Response: We thank the Reviewer for this comment. We modified the section and reads as follows:

“The reference value for P1NP is between 15-59µg/l and for CTX is <57ng/ml, respectively. However, patients treated with iron sucrose sucroferric oxyhydroxide had increased serum levels with 68.71 µg/l of P1NP and 0.60ng/ml for CTX beyond the reference values two weeks after iron substitution therapy.” (page 8, lines 281-284).

Reviewer 2 Report

1. The small sample size and the fact that it is a single-center study, which may limit the generalizability of the findings. Authors should discuss it in the discussion.

2. All tables in an academic paper should be represented using a three-line table.

3. The style of reference should be the same.

Author Response

Responses to the Reviewers

Manuscript nutrients-2421876: Impact of intravenous iron substitution on serum phosphate levels and bone turnover markers – An open-label pilot study

Dear Ladies and Gentlemen,

We are very grateful to the Editor and the Reviewers for their expertise and have addressed all comments and suggestions brought forward. Please find below the details of our revision.

The responses to the Reviewer comments are marked as Response in this document. Changes in the revised manuscript and in this document are marked in turquoise. Deleted text is marked as strikethrough in the revised manuscript and in this document. Changes not related to the Reviewers’ comments are only corrected in the revised manuscript. Since no bibliography was added to the Responses to the Reviewers, all citations were removed from this document. In the manuscript, citations and bibliography were updated. Page and line numbering is referred to the marked and revised manuscript. Please notice that the title of the manuscript was slightly adapted, as suggested by one Reviewer.

Reviewer #2

Comments and Suggestions for Authors

  1. The small sample size and the fact that it is a single-center study, which may limit the generalizability of the findings. Authors should discuss it in the discussion.

Response: We thank the Reviewer for this thoughtful comment. The small sample size was mentioned in the Limitation of the Discussion:

“There are several limitations to this study. Due to the small sample sizes with two lost to follow up in group 1, the results must be interpreted with caution. However, even with this small sample size, such highly significant differences in serum phosphate levels could be detected comparing both drugs.” (page 10, lines 360-363).

  1. All tables in an academic paper should be represented using a three-line table.
    Response: As suggested by the Reviewer, the tables were presented as three-line tables. However, we want to refer to the Editorial office for manuscript formatting, as every journal has his own guidelines for presentation.

  1. The style of reference should be the same.

Response: We thank the Reviewer for this valid comment. All references were reviewed and changed accordingly to be consistent throughout the whole manuscript.

Reviewer 3 Report

The authors examined association between intravenous iron substitution therapy on serum phosphate levels and bone turnover markers. They hypothesized that different iv iron supplementation leads to different longitudinal adaptation of serum phosphate levels. The results showed difference of two iv applicated iron substitutes but after twelve weeks no difference was observed. One disadvantage of examination is low number of examined pts but final results can justify it.

Author Response

Responses to the Reviewers

Manuscript nutrients-2421876: Impact of intravenous iron substitution on serum phosphate levels and bone turnover markers – An open-label pilot study

Dear Ladies and Gentlemen,

We are very grateful to the Editor and the Reviewers for their expertise and have addressed all comments and suggestions brought forward. Please find below the details of our revision.

The responses to the Reviewer comments are marked as Response in this document. Changes in the revised manuscript and in this document are marked in turquoise. Deleted text is marked as strikethrough in the revised manuscript and in this document. Changes not related to the Reviewers’ comments are only corrected in the revised manuscript. Since no bibliography was added to the Responses to the Reviewers, all citations were removed from this document. In the manuscript, citations and bibliography were updated. Page and line numbering is referred to the marked and revised manuscript. Please notice that the title of the manuscript was slightly adapted, as suggested by one Reviewer.

Reviewer #3

Comments and Suggestions for Authors:

The authors examined association between intravenous iron substitution therapy on serum phosphate levels and bone turnover markers. They hypothesized that different iv iron supplementation leads to different longitudinal adaptation of serum phosphate levels. The results showed difference of two iv applicated iron substitutes but after twelve weeks no difference was observed. One disadvantage of examination is low number of examined pts but final results can justify it.

Response: We thank the Reviewer for this positive assessment on the study.

Round 2

Reviewer 1 Report

The authors significantly improved the quality of their manuscript and addressed most comments. However, I would like to underline some remaining issues.

Major:

I feel uncomfortable to see a modification of the dosage of FC administered to the patients. Moreover, the lack of precision of the way they included their patients given the high similarity in the two groups is still odd.

To clarify it and close this topic, I think the authors should share the precise protocol registered by their ethic committee to clarify it (at least with the editor and me privately, or ideally as a supplemental material).

Minor

1.     In the abstract to improve the readability, please merge

a.     L26-27 “Two weeks after drug administration, significant differences were observed between 26 both study groups for phosphate (p<0.001) and ferritin (p<0.001)”

b.     AND L32-33 “. In group 1, phosphate levels were 32 lower and ferritin levels were higher compared to group 2”.

2.     In the method section :

L88-90: the authors stated they included patients with iron deficiency anemia but it is still highly unprecise: nutritional deficiency due to food restriction? Chronic blood losses in premenopausal women? Malabsorption due to bariatric surgery?

Please improve this point, it is crucial because it may have influence significantly the phosphate level variations.

Author Response

Responses to the Reviewers

Manuscript nutrients-2421876: Impact of intravenous iron substitution on serum phosphate levels and bone turnover markers – An open-label pilot study

Dear Ladies and Gentlemen,

We are very grateful to the Editor and the Reviewers for their expertise and have addressed all comments and suggestions brought forward. Please find below the details of our revision.

The responses to the Reviewer comments are marked as Response in this document. Changes in the revised manuscript and in this document are marked in turquoise. Deleted text is marked as strikethrough in the revised manuscript and in this document. Changes not related to the Reviewers’ comments are only corrected in the revised manuscript. Since no bibliography was added to the Responses to the Reviewers, all citations were removed from this document. In the manuscript, citations and bibliography were updated. Page and line numbering is referred to the marked and revised manuscript. Please notice that the title of the manuscript was slightly adapted, as suggested by one Reviewer.

Reviewer #1

Comments and Suggestions for Authors:

The authors significantly improved the quality of their manuscript and addressed most comments. However, I would like to underline some remaining issues.

I feel uncomfortable to see a modification of the dosage of FC administered to the patients. Moreover, the lack of precision of the way they included their patients given the high similarity in the two groups is still odd.

To clarify it and close this topic, I think the authors should share the precise protocol registered by their ethic committee to clarify it (at least with the editor and me privately, or ideally as a supplemental material).

Response: Again, we thank the Reviewer for his great interest in our study and his generally positive assessment of the study. We have attached the protocol for the ethics committee as well as the ethical approval. However, both documents are in German.

  1. In the abstract to improve the readability, please merge
  2. L26-27 “Two weeks after drug administration, significant differences were observed between both study groups for phosphate (p<0.001) and ferritin (p<0.001)”
  3. AND L32-33 “In group 1, phosphate levels were lower and ferritin levels were higher compared to group 2”.

Response: As suggested, both sentences were merged and the paragraph read as follows:

“Two weeks after drug administration, significant differences were observed between both study groups for phosphate levels were significantly lower (p<0.001) in group 1 and ferritin levels were significantly higher (p=0.011 p<0.001) in group 1. Phosphate levels (0.8-1.45mmol/L) were below the therapeutic threshold range and ferritin levels (10-200ng/mL for women and 30-300ng/mL for men) were above the therapeutic threshold range in group 1. P1NP (15-59µg/l) and CTX (<0.57ng/ml) were above the therapeutic threshold range in group 2. Four weeks after drug administration, significant differences were still observed between both study groups for phosphate (p=0.043) and ferritin (p=0.032 p=0.0009). In group 1, phosphate levels were lower and ferritin levels were higher compared to group 2.” (page 1, lines 26-34).

  1. In the method section: L88-90: the authors stated they included patients with iron deficiency anemia but it is still highly unprecise: nutritional deficiency due to food restriction? Chronic blood losses in premenopausal women? Malabsorption due to bariatric surgery?

Please improve this point, it is crucial because it may have influence significantly the phosphate level variations.

Response: We thank the Reviewer for this comment. Patients with nutritional deficiency due to food restriction, chronic blood losses in premenopausal women, or malabsorption due to bariatric surgery were not included in this study. This was added to the manuscript. Definitely, there are numerous causes for blood loss and anemia. However, as stated in the manuscript, solely patients with “an underlying chronic inflammatory bowel disease (Crohn’s Disease, ulcerative colitis) with occult blood loss determined by a fecal test or another underlying disease requiring intravenous iron substitution such as iron deficiency anemia” (page 2, lines 87-91) were included in this study. In our opinion, it is therefore not necessary to list all other reasons for blood loss or anemia as inclusion criteria are clearly mentioned.

Reviewer 2 Report

It can be accepted.

Author Response

Responses to the Reviewers

Manuscript nutrients-2421876: Impact of intravenous iron substitution on serum phosphate levels and bone turnover markers – An open-label pilot study

Dear Ladies and Gentlemen,

We are very grateful to the Editor and the Reviewers for their expertise and have addressed all comments and suggestions brought forward. Please find below the details of our revision.

The responses to the Reviewer comments are marked as Response in this document. Changes in the revised manuscript and in this document are marked in turquoise. Deleted text is marked as strikethrough in the revised manuscript and in this document. Changes not related to the Reviewers’ comments are only corrected in the revised manuscript. Since no bibliography was added to the Responses to the Reviewers, all citations were removed from this document. In the manuscript, citations and bibliography were updated. Page and line numbering is referred to the marked and revised manuscript. Please notice that the title of the manuscript was slightly adapted, as suggested by one Reviewer.

Reviewer #2

Comments and Suggestions for Authors:

It can be accepted.

Response: We thank the Reviewer for this positive assessment of the study.
